# Impact of Economic Accessibility on Realized Utilization of Home-Based Healthcare Services for the Older Adults in China

**DOI:** 10.3390/healthcare9020218

**Published:** 2021-02-17

**Authors:** Xiaodong Di, Lijian Wang, Liu Yang, Xiuliang Dai

**Affiliations:** School of Public Policy and Administration, Xi’an Jiaotong University, No 28 Xianning West Road, Xi’an 710049, China; dxd6794860@stu.xjtu.edu.cn (X.D.); yangliu777@stu.xjtu.edu.cn (L.Y.); xiuliangdai@stu.xjtu.edu.cn (X.D.)

**Keywords:** economic accessibility, realized utilization, family support, inverted U-shaped effect

## Abstract

Home-based healthcare service has gradually become the most important model to cope with aging in China. However, the contradiction between oversupply and insufficient demand of healthcare services is becoming increasingly serious. How to effectively improve the realized utilization of healthcare resources has become a key issue in the development of healthcare services. Based on the social background of “getting old before getting rich”, this article explores the relationship between economic accessibility and realized utilization, and finds that the impact of economic accessibility on realized utilization is inverted U-shaped, not a linear positive effect. In addition, considering the moderating role of family support, it is found that family support can strengthen the inverted U-shaped effect of economic accessibility on realized utilization. Therefore, exerting the role of family and improving economic accessibility can effectively solve the dilemma of low utilization of healthcare services.

## 1. Introduction

By the end of 2019, the Chinese population of 60 years old and above was close to 254 million people, accounting for 18.1% of the total population of China. China is facing a series of aging problems, such as labor shortage, insufficient pension, etc. Home-based healthcare services are some of the main healthcare models to cope with aging in China, including daily care, medical care, spiritual consolation, entertainment, etc., which is mainly supplied by the community center. It not only effectively meets the needs of the older adults to live at home, but also makes up the lack of family care. The central and local governments have also continued to upgrade the construction of the home-based healthcare service. By the end of 2019, there were 64,000 healthcare institutions and facilities, and 101,000 mutual-aid healthcare facilities, with a total of 3.362 million beds in the community. Although the construction of healthcare resources and service contents for the elderly is fast, and the coverage of the healthcare service is becoming increasingly extensive, the vacancy rate of the healthcare service facilities is as high as 30% or more. How to balance the increasingly sufficient supply and the insufficient demand of healthcare services? How to exert a valid role has become an urgent problem to healthcare services for older adults.

Under the economic and social background of “getting old before getting rich”, the older adults suffer from low earnings and a higher financial burden. One of the main barriers to accessing healthcare services is the high cost of healthcare visits. More than 60% of elderly people in China have no access to healthcare services due to poverty [1], so the economic accessibility of healthcare services has become a key factor restricting realized utilization [2]. Therefore, how does the economic accessibility affect realized utilization of healthcare services? Besides economic accessibility, what other factors affect the realized utilization of healthcare services? How can the realized utilization of healthcare services be effectively improved? These issues need further exploration.

In terms of the research literature, most studies on the utilization and economic accessibility of healthcare services focus on medical services and medical insurance [3,4,5], and believe that there is a positive correlation between income and utilization of healthcare services. Considering the publicity of healthcare services, this article uses the Anderson model as the theoretical basis, and puts forward the inverted U-shaped hypothesis about the realized utilization and economic accessibility of healthcare services and introduces the moderating variable of family support. Then, it explores the impact of the economic accessibility on realized utilization of healthcare services through empirical analysis and analyzes the economic reasons, which has certain guiding significance for improving economic accessibility and increasing the realized utilization of healthcare services.

## 2. Literature Review and Hypothesis

### 2.1. Realized Utilization

Referring to the definition of Andersen and Babette [6,7], the realized utilization of healthcare services refers to the use of public facilities and care services of home-based healthcare services for the older adults, affected by multiple systems and individual characteristics. In China, the realized utilization of healthcare services is very low, and the phenomenon of service surplus is generally serious, resulting in a large amount of idle resources, so the participation of the older adults is not ideal. Regarding the research on the realized utilization, most scholars use the Anderson model as the theoretical basis [8], and mainly focusing on the factors affecting the utilization of healthcare services. Liao used the Anderson model and found that marital status, family members and psychosocial factors play a very important role in elderly long-term care service needs in China [9]. Razak pointed out that older persons with different wealth status, living arrangement and social support, had a significant difference on utilizing health care from the perspective of Health Protection Scheme [10]. Bähler found that multimorbidity was associated with substantial higher healthcare utilization in Switzerland [11].

### 2.2. Economic Accessibility

Based on the definition of the World Health Organization, economic accessibilityis a measure of people’s ability to pay for services without financial hardship, and takes into account not only the price of the health services but also indirect and opportunity costs [12]. Itis related to attitude related to personal health risk, such as preventive savings and pension insurance, and mainlydepends on two key factors of the price of a commodity or service and income [13]. Firstly, healthcare services belong to a typical service industry with high input costs and long return periods. Cost of healthcare servicesare related to the cost of using health care facilities, the individual’s abilitytopay, perception of valueformoney, understanding of healthcare prices, total costs (direct and indirect) and possible health care financing arrangements [14]. Secondly, the economic accessibility of healthcare services is also related to people’s income [15]. Besides income sources, it also depends on other sources such as savings, borrowings and other mechanisms to finance the health needs [16]. Snowden found that the affordability of health care for households was significantly affected by income, as income increased, problems with affordability decreased [17]. Montu and Somdutta thought that the richest class of the rural sector had the highest utilization of public healthcare facilities and the poorest class had the lowest utilization [18]. Dimitris pointed out that higher income as an indicator of the disposable financial means of individuals directly increased healthcare affordability, while those with lower incomes often had restricted access to healthcare services [19].

### 2.3. Economic Accessibility and Realized Utilization

Economic accessibility is an important factor that affects the realized utilization of healthcare services, but its role has not yet reached a unified conclusion. Most research results show that, the older adults are sensitive to the cost of healthcare services when the economic accessibility is low, since the demand for healthcare services is mostly repetitive and long-term services. Lack of financial ability will cause families to sacrifice basic necessities of life, along with the severe financial burden on the households which in turn discourage them from seeking timely care [20]. Coupled with the traditional concept that Chinese elderly people pay more attention to saving, healthcare costs have become main factor affecting the realized utilization of healthcare services.

Most scholars believe that the older adults’ economic accessibility is higher when they possess more income, and the ability of the potential healthcare needs to be transformed into effective needs is greater, so the realized utilization of healthcare services is higher. Kumar analyzed data from the 2013 to 2014 National Health Interview Survey, and found lack of affordability adversely impacted accessibility of medical care [21]. Muhammad thought that higher percentage of healthcare-seeking patients received alternative healthcare because of the lower cost and easier access compared to modern healthcare [22]. Nel Jason analyzed the association between utilization and total health care costs in Philippines Health Reform, and found that the increase on health care costs limited utilization of health care though the reform decreased barriers to health care utilization [23].From the research results, the above existing literature believes that economic accessibility has a positive effect on realized utilization.

In China, the supply of healthcare services is at an early stage. The contents of healthcare services mostly focus on daily care, which are single and the quality is low. For basic healthcare service, older adults with different affordability have different using willing and realized utilization [24].Therefore, economic accessibility on realized utilization is not a linear positive effect. The article believes that, the cost of healthcare services is no longer the main factor affecting the utilization of healthcare services when the economic accessibility of healthcare services improves, because the expenditure on healthcare services accounts for a smaller and smaller proportion of the total economic expenditure. The older adults with higher affordability no longer pay attention to the economic accessibility of healthcare services, and instead pay more attention to the quality and professionalism of healthcare services. Therefore, the older adults with high incomes are likely to reduce their demand for basic healthcare services, thereby decrease the realized utilization of public healthcare services. Based on the above analysis, we can draw a conclusion that economic accessibility has a negative effect on realized utilization for high-income older adults.

According to Haans et al. [25], an inverted U-curve may be constructed by interacting two latent linear functions, one positive and one negative in the realized utilization.

**Hypothesis** **1** **(H1)**:
*The effect of economic accessibility on realized utilization of healthcare services for the older adults is inverted U-shaped.*


### 2.4. Family Support, Economic Accessibility and Realized Utilization

In addition to the economic accessibility that affects the realized utilization of healthcare services, other supports from family members such as financial support, life care, and spiritual comfort will also affect the utilization of healthcare services [26]. In China, it is a traditional virtue that families provide healthcare services for the elderly. However, with the development of industrialization, urbanization, and the normalization of population mobility, the function of family care for the elderly is gradually weakening, and family support is significantly different in the process of using community healthcare services for the elderly. Some scholars believe that if the older adults receive rich family support, family support can meet most of the healthcare services needed by the elderly, thereby reducing their dependence on community healthcare services [21]. Some scholars believe that, the more family support the elderly receive, the less pressure the elderly have on the cost of healthcare services. Additionally, they are more willing to reduce precautionary savings to meet their needs for diverse healthcare services, and increase the utilization of healthcare services. Zhang analyzed the changes in needs of healthcare services for the elderly in China over the period between 2005 and 2014, and found the lack of family supports created a serious barrier for them to seek healthcare services [27].

This article believes that family support, by influencing the older adults’ subjective perception of economic accessibility and reducing the economic barriers to using healthcare services [28], have a certain moderating effect on the realized utilization of healthcare services. When the economic accessibility of healthcare services is low, the more family support the elderly receive, the more sufficient information about healthcare services can be obtained, and the older adults have more opportunities to find, approach and use healthcare services. Especially among the poorer, inequalities between those who have and those who have not family support increase differences in realized utilization of healthcare services. With family support, the older adults are easier to reduce preventive savings and ignore the economic barriers in the use of healthcare services, thereby increasing the utilization frequency of healthcare services. When the economic accessibility of healthcare services is higher, the more family support the older adults receive, the more adequate healthcare resources they can use are. To meet the needs of special nursing, rehabilitation nursing and other healthcare services, the older adults are more willing to obtain high-quality healthcare services through professional channels, thereby reducing the demand for basic healthcare services in the community.

**Hypothesis** **2** **(H2)**:
*Family support strengthens the inverted U-shaped effect of the economic accessibility on realized accessibility of healthcare services for the older adults.*


In summary, the relationship among family support, economic accessibility and realized utilization of healthcare services is shown in Figure 1, and the conceptual framework of this article is shown in Figure 2.

## 3. Data and Variables

### 3.1. Data Resource

The data come from a summer survey of major national projects in 2019. The survey team is composed of more than 20 members of teachers, doctoral candidates and master’s students from Xi’an Jiaotong University. These members have professional backgrounds in healthcare services and have certain investigative capabilities. Since the remuneration of a valid questionnaire is RMB 30, under the premise of unified training, every member carefully fills out the questionnaire in form of self-administered questionnaires, which ensuring the quality of the questionnaire to a certain extent. This survey is aimed at the accessibility of healthcare services in Shaanxi Province, China. The status quo was investigated by stratified sampling. The first stage of stratification is based on cities, the second stage of stratification is based on counties, the third stage of stratification is based on townships, and the fourth stage of stratification is based on villages. The fifth stage of stratification takes the elderly as a sample. Finally, we investigate three representative regions of Hanzhong, Baoji, and Yan’an, which are characterized by high aging and rapid development of healthcare service for the older adults. The aging characteristics of these areas are similar to the healthcare services in most areas in China. Finally, the team obtained a total of 948 valid questionnaires, and we adopt 832 valid questionnaires related to the theme, by discarding missing data and invalid questionnaires. Additionally, the sample’s reliability value of Cronbach’s α is 0.77, which confirms that the included variables are appropriate for calculating a further analysis.

### 3.2. Variable Description

This article uses realized utilization as the dependent variable. The realized utilization of healthcare services can be measured by indicators such as willingness, satisfaction, frequency of use, and facilities occupancy rate [10,29]. Therefore, we adopt the question of “Do you often use the healthcare services provided by the community” to measure the realized utilization of healthcare services. The higher the utilization frequency is, the higher the utilization rate of community healthcare resources is.

This article takes economic accessibility as the independent variable. Economic accessibility can be measured by indicators such as elderly income, savings, and healthcare costs [30]. Considering the different contents of healthcare services, we adopt four questions of “Can you afford the corresponding living care service costs”, “Can you afford the corresponding medical and nursing service costs”, “Can you afford the corresponding entertainment service costs” and “Can you afford the elderly rights protection services provided by the community” to measure the economic accessibility of healthcare services. According to the average segmentation method, the total score is the sum of the scores of the four questions, and the total score of 4–6 is recorded as 1, 7–10 is recorded as 2, 11–14 is recorded as 3, and 15–17 is recorded as 4, 18–20 is recoded as 5. The higher the total score, the higher the economic accessibility of healthcare services for the older adults.

This article takes family support as the moderating variable. Family support can be measured by indicators such as children’s financial support, number of children, family members and understanding from family members [31]. Considering different types of family support that affect economic accessibility, we choose the question of “How many people are in your family” to measure family support. The larger the number of family members is, the more family supports of economic power and spiritual support the elderly can rely on are, and the older adults have less pressure and more opportunities to use healthcare services.

The realized utilization of healthcare services is affected by various factors such as individual characteristics and consumption environment. In terms of individual characteristics, age, income, education level and health status will all affect the perceived affordability of the older adults [31,32,33]. Especially, some factors of the conservative consumption concepts, strong savings motivation and weak information processing capabilities can significantly reduce the willingness of the older adults to use healthcare services. In terms of the consumption environment, the imbalanced interaction between market suppliers, imperfect service supply system, and asymmetry of consumption information have severely increased the actual burden of older adults to use healthcare services, which results in an imbalance between the supply and demand and low realized utilization of healthcare services. Therefore, we take age, education level, psychosocial status, residence, self-assessed health status and gender as control variables [17,21], which can ensure that the results are not affected by individual characteristics. Among them, the psychosocial status refers to the attitudes and emotions of the elderly in facing different physical conditions, economic conditions, and life situations [34]. Besides, for the two categorical variables of residence and gender, we set as dummy variables with male = 1, female = 0, and urban = 1, rural = 0.

The indicators assessing method adopts the Likert 5-point scale method, and assigns values of “1, 2, 3, 4, 5” from weak to strong. The basic situation of the sample is shown in Table 1.

It can be seen from Table 2 that the standard deviation is close to 1, indicating that the data is concentrated near a certain central value, and the sample has no outliers. In terms of economic accessibility, its mean and median are 3.65 and 4respectively, indicating that the older adults in the sample have a relatively strong affordability for the cost of healthcare services, which shows that the cost of healthcare services at this stage is relatively low. In terms of realized utilization, the mean and median are 2.99 and 3, respectively, indicating that the frequency of realized utilization of healthcare services has not reached a high level and the older adults’ awareness of using community healthcare services has not yet formed. Overall, the sample conforms to the actual situation of the older adults and has certain reliability.

## 4. Results and Discussion

This article mainly focuses on the impact of economic accessibility on realized utilization of healthcare services. The dependent variable is divided into five categories from weak to strong, which belongs to ordered variables, so it is suitable to adopt ordered logit regression models. The basic form is as follows.
(1)$$P(y>j/x_{i})=\frac{\exp(\alpha +\beta x_{i})}{1+\exp(\alpha +\beta x_{i})}$$

In the Formula (1), y represents independent variable,y=1,2,3,4,5. $x_{i}$ represents the *i*-th factor that affects independent variable.

### 4.1. Accessibility on Realized Utilization

In order to verify Hypothesis 1, we take economic accessibility as the independent variable, realized utilization as the dependent variable, age, education level, psychosocial status, residence, health status and gender as the control variables, and construct Formula (2). The results are shown in Table 2.
(2)$$\log\mathrm{it}\left(P(y>j|x\right)=\alpha_{1}+\beta_{1}X^{2}+\gamma_{1}X+\varphi_{1}Z,j=1,2,3,4,5$$

In the Formula (1): X represents independent variable.Z represents control variable. $\alpha_{1}$, $\beta_{1}$, $\gamma_{1}$, $\varphi_{1}$ represents regression coefficients.

In Model1–6, the control variables are successively added to control the stability of the model results. According to Table 2, the squared economic accessibility has a negative impact on realized utilization, and economic accessibility has a positive impact on realized utilization. The results are stable with the introduction of different control variables, and are significant at 1% level. It passes the log likelihood test. Therefore, the impact of economic accessibility on realized utilization first increase and then decrease, showing an inverted U-shaped trend. Hypothesis 1 is supported. With the development of healthcare services, the older adults can afford more types of healthcare services when the economic cost is reduced to a certain range, and they will increase the utilization of basic healthcare services to meet their basic needs. When the economic accessibility increases continuously, the older adults’ demand will decline due to the low demand elasticity of basic healthcare services, and the realized utilization of healthcare services will also decline.

In terms of the control variables, education level has a significant impact of −0.2039 on realized utilization, indicating that the higher the education level is, the lower realized utilization is. The main reason is the older adults will possess higher income when their education level is high, and they are more willing to focus on the quality of healthcare services. The psychosocial status has a significant impact of 0.2935 on realized utilization, indicating that the older adults are more willing to use healthcare services when their psychosocial status is positive, which is mainly related to the older adults’ ability to perceive and find healthcare services. The impact of residence on realized utilization is −0.2537 and it is significant, indicating that urban elderly people are not dependent on community healthcare services and utilization frequency is lower. The possible reason is that the urban older adults have more opportunities to choose different types of healthcare services. Age has a significant impact of 0.0851on realized utilization, indicating that the older the elderly is, the higher the frequency of using healthcare services is. The main reason is that the self-service ability of older adults will decline due to the increase of age, which increases the demand for community healthcare services. However, health status and gender have no significant impact on the realized utilization, indicating that the differences in gender and health status are non-significant after controlling for other socioeconomic determinants of utilization.

### 4.2. The Moderating Role of Family Support

In order to verify Hypothesis 2, we take family support as the moderating variable, age, education level, psychosocial status, residence, health status and gender as the control variables, test the moderating role of family support in the impact of economic accessibility on realized utilization, and construct Formula (3). The results are shown in Table 3.
(3)$$\log\mathrm{it}\left(P(y>j|x\right)=\alpha_{2}+\beta_{2}X^{2}+\gamma_{2}X+\mu X^{2}\times M+\lambda X\times M+\varphi_{2}Z,j=1,2,3,4,5$$

In the Formula (2): X represents independent variable.Z represents control variable. M represents moderating variable.$\alpha_{2}$, $\beta_{2}$, $\gamma_{2}$, μ, λ, $\varphi_{2}$ represents regression coefficients.

In Models 7–12, we test the impact of the cross item of family support and squared economic accessibility on realized utilization when the control variables are successively added to control the stability of the model results. According to Table 3, the final impact result is 0.0266, and the impact of the cross item of family support and economic access on realized utilization is −0.1557. In Model 13, under the interactions of the family support variable with all control variables, the impact of the cross item of family support and squared economic accessibility on realized utilization is 0.0593. The results are stable with the introduction of different cross item and are significant at 1% level, indicating that the family support has strengthened the inverted U-shaped effect of the economic accessibility on realized utilization of healthcare services. Hypothesis 2 is supported. When economic accessibility is low, family support cannot only reduce the cost pressure of healthcare services, but also mentally reduce the perceived pressure of using healthcare services and increase the realized utilization. When economic accessibility is high, higher family support reduces the older adults’ demand for basic healthcare services. The older adults pay more attention to high-quality and personalized healthcare services, no longer rely on community healthcare services, and reduce the utilization of healthcare services.

### 4.3. Discussion

From the perspective of the impact of economic accessibility on realized utilization, the realized utilization of healthcare services will increase when the economic accessibility is low. As economic accessibility increases to a certain level, the realized utilization of healthcare service will decrease. Therefore, we put forward the following three suggestions to effectively improve the realized utilization. This suggestion is applicable to developing countries where healthcare services for the older adults are at an early stage.

Firstly, the responsibility of the service subject should be clarified to encourage the utilization of healthcare services, and different economic strategies should be applied to different groups. For basic healthcare services, the target group is mainly low- and middle-income older adults. The government would provide fairly effective financial protection to ensure the equitable utilization of basic healthcare. For high-quality and professional healthcare services, the target group is mainly high-income older adults. The healthcare services should be led by market, focus on diversified healthcare contents and cultural and entertainment projects.

Secondly, for low-income groups, the costs of healthcare services of primary care and daily nursing should be curtailed, which was consistent with the study of Zhu [30]. Intergenerational communication, social networks, and media promotion can be used to change the consumption awareness and consumption habits of the older adults, which will reduce the perceived burden of the older adults, and improve the realized utilization of healthcare services. Besides, the development of a social health insurance scheme should be a long-term goal to reduce older adults’ financial burden and increase the utilization of healthcare services in China [35].

Thirdly, for high-income groups, healthcare costs are no longer the main reason affecting realized utilization of healthcare services for older adults. The realized utilization of healthcare service will depend on contents and quality of healthcare service. In order to attract high-income groups to use community healthcare services, the supply of healthcare services should pay attention to diversification and high quality. The consumption market for healthcare services should be opened through the development of door-to-door services or professional healthcare services. Suppliers should set prices at different levels to meet the different needs of older adults, which will promote the consumption structure of healthcare services, and realize the upgrade from basic to quality healthcare services. In addition, we should remove the information barriers for the older adults to use healthcare services, and make it easier for the older adults to obtain safe and accurate consumption information.

From the perspective of family support, family support can affect the realized utilization by affecting economic accessibility. The impact includes two parts of efficiency and equity. In efficiency, we should focus on the positive effect of the family support, and encourage family members to provide mental support and financial support. Family members can reduce information barriers and facilities barriers to use healthcare services for the older adults through intergenerational support, by shifting the burden of caring for the elderly to their family for the government, so the Government needs to take measures such as service subsidies and skill training to encourage family members to provide some healthcare services. In equity, for the older adults with high family support, the economic accessibility is higher than those who have low family support, so some policies such as personal income tax reduction and exemption can encourage family members to provide more financial support to undertake maintenance obligations. Thereby, family support reduces public expenditure, but increases socioeconomic inequalities in utilization.

## 5. Conclusion and Limitations

### 5.1. Conclusion

Based on a literature review and theoretical analysis, this article puts forward an inverted U-shaped hypothesis about the economic accessibility and realized utilization of healthcare services. Then, taking into account the role of family support, we empirically test the relationship among family support, economic accessibility and realized utilization. Finally, we draw the following conclusions.

Firstly, economic accessibility has an inverse U-shaped effect on realized utilization. When the economic accessibility is low, the realized utilization of healthcare services will increase with the improvement of the older adults’ affordability. When the affordability reaches a certain range, economic accessibility is no longer the main factor affecting realized utilization, and the older adults’ reliance on community healthcare services will gradually decline.

Secondly, family support strengthens the inverse U-shaped effect of economic accessibility on realized utilization. Family support can weaken the economic barriers for low-income older adults to use healthcare services, and at the same time accelerate high-income older adults to get rid of their dependence on basic healthcare services.

Thirdly, in terms of economic accessibility, the prices of healthcare services shall be set at different levels to reduce the cost of basic healthcare services. Government should develop a social health insurance scheme as a long-term goal to reduce older adults’ financial burden. In terms of family support, intergenerational support can be used to reduce economic barriers to use healthcare services and improve the realized utilization of healthcare services for the older adults.

### 5.2. Limitations

Although considering the impact of economic accessibility on realized utilization of healthcare services, this article does not cover all the contents of healthcare services, such as spiritual comfort and emergency treatment. In addition to the factors of economic accessibility and family support, service policies, social organization, personal preferences and social fairness also affect the utilization of community healthcare services. Therefore, follow-up research on the utilization of healthcare services can also include evaluation of efficiency and equity for resource utilization, the impact of policy mix on the utilization, and the effect of social organizations on utilization in healthcare services. In addition, we can also discuss the impact of economic accessibility on realized utilization with the effect of COVID-19.

## Figures and Tables

**Figure 1 healthcare-09-00218-f001:**
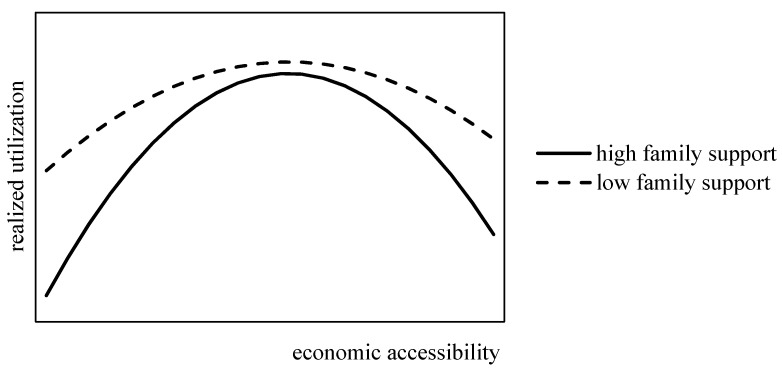
The relationship among family support, economic accessibility and realized utilization of healthcare services.

**Figure 2 healthcare-09-00218-f002:**
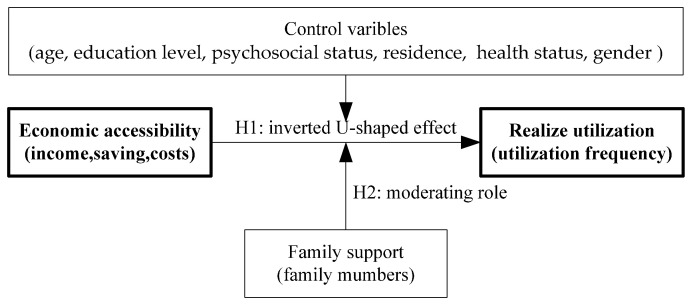
The conceptual framework and hypothesis.

**Table 1 healthcare-09-00218-t001:** Variables, indicators and sample characteristics.

Variable	Indicator	Item	Options (Data)	Frequency	Percentage (%)	Mean	Median	Standard Deviation
Dependent variable	Realized utilization	The utilization frequency of healthcare services	① seldom(1) ② sometimes(2) ③ often (3) ④ usually(4) ⑤ always(5)	56 243 270 183 80	6.73 29.21 32.45 21.99 9.62	2.99	3	1.08
Independent variable	Economic accessibility	Affordability of healthcare services costs	① totally not(1) ② partly not(2) ③ can(3) ④ most(4) ⑤ totally(5)	226 161 159 232 54	27.16 19.35 19.11 27.89 6.49	3.65	4	1.21
Moderating variable	Family support	Family members	quantity			2.67	2	1.59
Control variable	Individual characteristics	Gender	① male (1) ② female (0)	346 486	41.59 58.41			
		Education level	① Elementary (1) school and below ② Junior high school(2) ③ high school(3) ④ college(4) ⑤ bachelor degree and above (5)	373 226 164 48 21	44.83 27.17 19.71 5.77 2.52			
		Psychosocial status	① very bad(1) ② bad (2) ③ neither good, nor bad (3) ④ good (4) ⑤ very good (5)	6 51 91 303 381	0.72 6.13 10.94 36.42 45.79	4.20	4	0.92
		Residence	① urban (1) ② rural (0)	472 360	56.73 43.27			
		Health status	① very bad(1) ② bad (2) ③ neither good, nor bad (3) ④ good (4) ⑤ very good(5)	39 153 227 278 134	4.69 18.39 27.40 33.41 16.11	3.38	3	1.10
		Age	① 60–64 (1) ② 65–69 (2) ③ 70–74 (3) ④ 75–79 (4) ⑤ 80 and above (5)	164 193 172 127 176	19.71 23.20 20.67 15.27 21.15			

**Table 2 healthcare-09-00218-t002:** Impact of economic accessibility on realized utilization.

	Model 1	Model 2	Model 3	Model 4	Model 5	Model 6
1- squared	−0.0721 * (−1.8441)	−0.0699 * (−1.7848)	−0.0720 * (−1.8346)	−0.0699 * (−1.7823)	−0.0717 * (−1.8261)	−0.0677 * (−1.7208)
economic accessibility	0.5767 ** (2.2655)	0.5903 ** (2.3149)	0.5448 ** (2.1296)	0.5358 ** (2.0952)	0.5447 ** (2.1279)	0.5352 ** (2.0915)
gender	0.1190 (0.9418)	0.2275 * (1.7518)	0.1976 (1.5184)	0.1825 (1.3995)	0.1737 (1.3279)	0.1691 (1.2918)
education level		−0.2328 *** (−3.7377)	−0.2453 *** (−3.9377)	−0.2045 *** (−3.0451)	−0.2120 *** (−3.1281)	−0.2039 *** (−3.0021)
psychosocial status			0.3214 *** (4.5541)	0.3261 *** (4.6127)	0.3081 *** (4.1870)	0.2935 *** (3.9624)
residence				−0.2228 (−1.6250)	−0.2278 * (−1.6612)	−0.2537 * (−1.8395)
health status					0.0543 (0.8849)	0.0788 (1.2569)
age						0.0851 * (1.8716)
cut1	−1.5637 *** (−3.9130)	−1.9115*** (−4.6472)	−0.8414** (−1.7814)	−0.8818* (−1.8660)	−0.7914 (−1.6363)	−0.4993 (−0.9828)
cut2	0.5021 (1.2921)	0.1744 (0.4371)	1.2704 *** (2.7250)	1.2348 *** (2.6482)	1.3236 *** (2.7733)	1.6210 *** (3.2227)
cut3	1.8654 *** (4.7270)	1.5517 ** (3.8433)	2.6774 *** (5.6404)	2.6453 *** (5.5730)	2.7354 *** (5.6299)	3.0392 *** (5.9293)
cut 4	3.3402 *** (8.2046)	3.0361 *** (7.3066)	4.1831 *** (8.5784)	4.1527 *** (8.5168)	4.2457 *** (8.5039)	4.5514 *** (8.6552)
log likelihood	−1213.74	−1206.69	−1196.19	−1194.87	−1194.48	−1192.73
LR	9.65 **	23.67 ***	44.66 ***	47.30 ***	48.09 ***	51.59 ***

Note: T-values are in parentheses. *** means significant at the 1% level, ** means significant at the 5% level, and * means significant at the 10% level.

**Table 3 healthcare-09-00218-t003:** The moderating role of family support.

	Model 7	Model 8	Model 9	Model 10	Model 11	Model 12	Model 13
economic accessibility(squared)	−0.1412 *** (−2.6179)	−0.1321 ** (−2.4531)	−0.1456 *** (−2.6961)	−0.1388 ** (−2.5617)	−0.1444 *** (−2.6580)	−0.1395 ** (−2.5643)	−0.2321 *** (−3.8797)
economic accessibility	0.9848 *** (3.2557)	0.9672 *** (3.2044)	0.9749 *** (3.2200)	0.9455 *** (3.1200)	0.9753 *** (3.2094)	0.9597 *** (3.1589)	1.6330 *** (4.6535)
family members×economic accessibility(squared)	0.0259 ** (1.9984)	0.0238 * (1.8381)	0.0276 ** (2.1269)	0.0260 ** (1.9959)	0.0270 ** (2.0697)	0.0266 ** (2.0419)	0.0593 *** (3.6903)
family members×economic accessibility	−0.1486 *** (−2.6752)	−0.1407 ** (−2.5278)	−0.1586 *** (−2.8448)	−0.1524 *** (−2.7239)	−0.1580 *** (−2.8155)	−0.1557 *** (−2.7798)	−0.3911 *** (−4.4584)
gender	0.1624 (1.2769)	0.2702 ** (2.0678)	0.2390 * (1.8259)	0.2260 * (1.7234)	0.2135 (1.6230)	0.2093 (1.5906)	0.0222 (0.5296)
education level		−0.2320 *** (−3.7091)	−0.2437 *** (−3.8941)	−0.2042 *** (−3.0300)	−0.2158 *** (−3.1706)	−0.2077 *** (−3.0450)	−0.0305 (−1.3707)
psychosocial status			0.3420 *** (4.8255)	0.3461 *** (4.8766)	0.3186 *** (4.3223)	0.3053 *** (4.1160)	0.0599 ** (2.5125)
residence				−0.2171 (−1.5781)	−0.2256 (−1.6389)	−0.2503 * (−1.8082)	−0.0814 * (−1.8505)
health status					0.0853 (1.3819)	0.1077 * (1.7066)	0.0400 * (1.9148)
age						0.0790 * (1.7313)	0.0307 ** (2.0870)
cut 1	−1.5387 *** (−3.8366)	−1.8959 *** (−4.5853)	−0.7427 ** (−1.5612)	−0.7889 * (−1.6569)	−0.6418 (−1.3146)	−0.3713 (−0.7246)	−0.6485 (−1.3665)
cut 2	0.5476 (1.4049)	0.2082 (0.5193)	1.3897 *** (2.9594)	1.3475 *** (2.8679)	1.4929 *** (3.0974)	1.7677 *** (3.4833)	1.4695 *** (3.1356)
cut 3	1.9368 *** (4.8881)	1.6116 *** (3.9698)	2.8267 *** (5.9054)	2.7878 *** (5.8224)	2.9362 *** (5.9755)	3.2166 *** (6.2131)	2.8881 *** (6.0420)
cut 4	3.4287 *** (8.3813)	3.1158 *** (7.4538)	4.3558 *** (8.8495)	4.3190 *** (8.7728)	4.4726 *** (8.8476)	4.7544 *** (8.9429)	4.3987 *** (8.9516)
log likelihood	−1204.70	−1197.76	−1185.96	−1184.72	−1183.76	−1182.26	−1193.62
LR	27.65 ***	41.53 ***	65.12 ***	67.62 ***	69.53 ***	72.53 ***	49.80 ***

Note: T-values are in parentheses. *** means significant at the 1% level, ** means significant at the 5% level, and * means significant at the 10% level.

## Data Availability

The data presented in this study are available.

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
