# Peer review of "Impact of Economic Accessibility on Realized Utilization of Home-Based Healthcare Services for the Older Adults in China"

_healthcare, 2021, doi:10.3390/healthcare9020218_

Round 1

Reviewer 1 Report

I find your paper interesting and stimulating and I appreciate it. Also the research questions are clear and the methodology well described. To improve the paper i suggest:

  1. In the introduction it could be useful to clarify the meaning of realised utilization. I don't know if it is only a problem related to the word "realized", anyway I would like to be sure that you mean actual cosumption. If I well understand you consider consumtion in terms of health expenditure. Do you have any evidence about the quality of health services you refer to?
  2. It is clear to me the economic accessibilty. I wonder if you can add something about health risk some other consideration about attitude related to personal health risk. of course it is not possible to consider it in the survey but you can add some comments in the framework design.
  3. You define "realized utilization" as the use of health services, satisfaction and frequency of utilization, but you do not clarify if the three dimensions have the same weight. I think it is better to clarify.
  4. In the conclusion, it would be useful to discuss if the behavior is related to chinese culture or if it could be generalized. You can propose other developments of the research in terms of the replication of the survey in other countries or culture contexts
  5. I do not see any consideration about the limits of the research. You can add some comments on them.

Author Response

1. In the introduction it could be useful to clarify the meaning of realised utilization. I don't know if it is only a problem related to the word "realized", anyway I would like to be sure that you mean actual consumption. If I well understand you consider consumption in terms of health expenditure. Do you have any evidence about the quality of health services you refer to?

Answer:Realised is a problem related to the word "realized". This article does not involve the quality of health services. Although quality is an important factor affecting the utilization of healthcare services, this article only focuses on the impact of economic accessibility on the utilization.

2. It is clear to me the economic accessibility. I wonder if you can add something about health risk some other consideration about attitude related to personal health risk. of course it is not possible to consider it in the survey but you can add some comments in the framework design.

Answer:Increase a sentence that economic accessibility is related to attitude related to personal health risk, such as preventive savings and pension insurance.

The attitude related to personal health risk is related to individual characteristic. Although this article does not directly measure attitude, it controls personal characteristics.

3. You define "realized utilization" as the use of health services, satisfaction and frequency of utilization, but you do not clarify if the three dimensions have the same weight. I think it is better to clarify.

Answer:This article does not adopt three dimensions, but only uses frequency of utilization to measure realized utilization. Although the measurement dimension is single, frequency can effectively reflect the degree of participation in healthcare services for older adults.

4. In the conclusion, it would be useful to discuss if the behavior is related to Chinese culture or if it could be generalized. You can propose other developments of the research in terms of the replication of the survey in other countries or culture contexts.

Answer:This article selects three research areas with high aging and rapid development of healthcare service for the older adults. The research results are applicable to most areas of China and some developing countries. Subsequent research can take other countries as research objects, and propose more meaningful research conclusions by comparative studies.

5. I do not see any consideration about the limits of the research. You can add some comments on them.

Answer: Increase limitations section.

thanks for your suggestion!

Reviewer 2 Report

Article 1 The Inverted U-shaped Effect of Economic Accessibility on Realized Utilization of Home-Based Healthcare Services for theOlder Adults in China: the Moderating Role of Family Support

This paper focuses on Home-based healthcare service for the elderly in China and the contradiction between oversupply and insufficient demand of healthcare services. The paper deals with an important issue: the insufficient demand for elderly healthcare in China due to poverty.
The paper considers the relationship between economic accessibility and utilization, and finds that the impact of economic accessibility on utilization is inverted U-shaped. The moderating role of family support is included in the analysis.
The Authors conclude that "family support can strengthen the inverted U-shaped effect of economic accessibility on utilization. Therefore, exerting the role of family and improving economic accessibility can effectively solve the dilemma of low utilization of healthcare services."(rows 18-20)

My opinion is that the paper deals with an important issue and it is original. However,  in order to be published the following revisions are needed in my opinion.

1- A main point is related to the language that is not always very clear ( see point 2 below)

2- There is the need of a better definition of some of the terms used:

Row 41 and other parts of the manuscript: The Authors use the term "realized utilization", it is not crear the definition of what you mean by this, I wonder if you mean " observed utilization of public community healthcare services" in practice, but, as this is a dependent variable in the estimated models, a theoretical definition / references are needed in the text e.g. at the beginning of par. 3.2., before the description of how it is measured (row n. 167)

Row 146: "Family support strengthens the inverted U-shaped effect of the economic accessibility on realized accessibility of healthcare services for the older adults."
I would add: while increasing inequalities in access and use.
There appear to be a trade-off here between efficiency ( shifting the burden of caring for the elderly to their family can be cost saving for the government) and equity (looking at those who do have a family support the economic accessibility gap is higher between the poorer and the richer than considering those who have low family support) (Figure 1)
Thereby, increasingly having people counting on their family would a) reduce public expenditure (increase in efficiency) but 2) increase socioeconomic inequalities in utilization (with a horizontal equity in utilization reduction). The equity implications need to be discussed in the discussion section.

The Authors use the term "economic accessibility of healthcare services". I would suggest to use the term "affordability" which seems more appropriate.

Row n. 16 "is inverted U-shaped, not a continuous promotion." Not clear what the Authors mean (maybe substitute "promotion" with "one"?)
Row n. 18 "exerting 18 the role of family". Not clear what the Authors mean (e.g. supporting families with public policies? incentives to family to take care of the elderly?)

Row 25. "spiritual consolation" or "spiritual comfort". Not clear what the Authors mean. Please explain more in detail.

Row 108 "the types of healthcare services are single" Not clear what the Authors mean.Please explain more in detail.

Row 116 "realized utilization of healthcare services" Again, not clear what the Authors mean, maybe "utilization of public healthcare services"? I guess the richer older adults may show relatively reduced demand for private healthcare , while increasing demand for private services in China.

Row 142 "To meet the needs of diverse healthcare services" Not clear what it is meant by "diverse" and the overall text.

Row 132-134
"This article believes that family support, by influencing the older adults' subjective perception of economic accessibility and reducing the economic barriers to using healthcare services [23], have a certain moderating effect on the realized utilization of healthcare services."

Not clear what the Authors mean by "older adults' subjective perception of economic accessibility". Do you mean the subjective feeling of economic and financial insecurity?

Row 211- "In terms of realized utilization, the mean and median are 2.99 and 3 respec- 211 tively, indicating that the realized utilization of healthcare services is average and the consumption market for healthcare services has not yet formed." Please explain.

Table 3- please substitute the variable names in the first column for better clarity

Most literature use Dummies for modelling the different educational levels . Please define them and report estimates, specifying the omitted one. The same for residence, which is the omitted one? It seems Urban but again, please specify the omitted in the variable definition section.

Row 256 : "However, health status and gender have no significant impact on the realized utilization, indicating that the impact of health status and gender on the realized utilization has not yet shown." Not clear. Please explain the last sentence. Maybe the system is equitable and differences in gender and health status are non significant after controlling for other socioeconomic determinants of utilization.

Please provide a definition of the health status variable and how the Likert scale is derived.Is it self-assessed health, or the n. of chronic conditions or objective health status?

3- ESTIMATION

Results in Table 3 seem to come from a simple logistic model not from a multinomial logit (you seem to have more than one outcome if I understood clearly). Can you please explain more the choice of the modelling strategy?
Overall, if I understood that you used a simple logistic model for the probability of outcome , it seems more appropriate to use ordered logit models as you seem to have an ordered dependent variable.

The effect of the interactions of the family support variable with all determinants as a final model including all interactions need to be added for completeness.

4-DISCUSSION

Row 290 "For high-income elderly groups, healthcare 290 costs are no longer the main reason affecting realized utilization of healthcare services." This is an expected and well known results in the empirical literature on the demand for health and healthcare services, so I would say that this confirms existing literature findings.

Rows 290-324
I would suggest to not go beyond the results obtained and cut/ heavily reduce this part discussion.
For example: row 292 : "The supply of healthcare services should pay attention to diversification and high quality. The consumption market for healthcare services should be opened through the development of door-to-door services or professional healthcare services. This suggestion is applicable to developing countries where healthcare services for the older adults are at an early stage"
This has no link with the role of family as a mediator in healthcare utilization and the study do not give any evidence from the supply side.

For example, from row 311:
"From the perspective of the demand side of healthcare services for the older adults, in order to improve the economic accessibility of healthcare services, we should also encourage the older adults to re-employed, invest financial project, buy commercial insurance, and delay retirement, in addition to increasing the coverage and protection of basic 314 old-age insurance and medical insurance. These actions are beneficial to realize the simultaneous growth of the income of the older adults and economic development. For low- income older adults, intergenerational communication, social networks, and media pro motion can be used to change the consumption awareness and consumption habits of the older adults, which will reduce the perceived burden of the older adults, and improve the realized utilization of healthcare services. For high-income older adults, we should remove the information barriers for the older adults to use healthcare services, and make it easier for the older adults to obtain safe and accurate consumption information. "

There are here a series of value judgemental considerations that are not linked to the results that cannot be supported by the research done.

I do not see why only for high income and not for the poorer it is important to "...make it easier for the older adults to obtain safe and accurate consumption information."
and the following statement:
"Finally, our aims are to transform potential needs into realized consumption behaviors, enlarge consumption market and upgrade consumption structure.(row 322)" is not linked with the paper aims and results.

5-Finally:

Row: 311: "From the perspective of the demand side of healthcare services for the older adults, in order to improve the economic accessibility of healthcare services, we should also encourage the older adults to re-employed, invest financial project, buy commercial insurance, and delay retirement, in addition to increasing the coverage and protection of basic 314 old-age insurance and medical insurance."
And :
Row n. 352: "Government should encourage the older adults to increase the income and reduce the economic barriers."

I think this cannot be done with the elderly if they are sick and frail. No insurance will sell packages to the over 65. In Europe Welfare systems allow the elderly to retire and to choose if they want to continue to work or not, because it is well known that retirement is a right and that it can be necessary for several reasons like, e.g., due to worsening healthcare conditions and not only for this reason. Seeing elderly sick people working to keep up income is not a great society achievement.
So I would strongly suggest to make the statements less general and more specific, e.g., giving incentives to the healthy retired elderly population on a voluntary - not compulsory basis- to go back to work for periods/part-time etc..

7-Minor remarks:

Typos:

Row 164 "Table1.", substitute with "Table 1."

Author Response

1. The Authors use the term "realized utilization", it is not clear the definition of what you mean by this, I wonder if you mean "observed utilization of public community healthcare services" in practice.

Answer:The realized utilization of healthcare services refers to the use of public facilities and care services of home-based healthcare services for the older adults.

2. "Family support strengthens the inverted U-shaped effect of the economic accessibility on realized accessibility of healthcare services for the older adults."
I would add: while increasing inequalities in access and use.

Answer:Family support not only strengthens the rise in the use of utilization when economic accessibility is insufficient, but also accelerates the decline in the use of healthcare services when economic access is sufficient.

3. There appear to be a trade-off here between efficiency and equity. The equity implications need to be discussed in the discussion section.

Answer:Although fairness is important factor affecting service utilization, it is not the focus of this article. Fairness is the focus of subsequent research.

4. The Authors use the term "economic accessibility of healthcare services". I would suggest to use the term "affordability" which seems more appropriate.

Answer:Affordability emphasizes personal income, while economic accessibility includes both personal income and the cost of services.

5. "is inverted U-shaped, not a continuous promotion." Not clear what the Authors mean (maybe substitute "promotion" with "one"?)

Answer:Change continuous promotion to linear positive effect.

6. "exerting the role of family". Not clear what the Authors mean (e.g. supporting families with public policies? incentives to family to take care of the elderly?)

Answer:The role of family refers to the healthcare services for the elderly provided by the family.

7. "spiritual consolation" or "spiritual comfort". Not clear what the Authors mean. Please explain more in detail.

Answer:Spiritual consolation refers to the mental needs and spiritual entertainment of the elderly.

8. Row 108 "the types of healthcare services are single" Not clear what the Authors mean. Please explain more in detail.

Answer:Change types to contents.

9. "realized utilization of healthcare services" Again, not clear what the Authors mean, maybe "utilization of public healthcare services"?

Answer:Change healthcare services to public healthcare services.

10. "To meet the needs of diverse healthcare services" Not clear what it is meant by "diverse" and the overall text.

Answer:Increase the contents of special nursing, rehabilitation nursing and so on to explain the diverse healthcare services.

11. Not clear what the Authors mean by "older adults' subjective perception of economic accessibility". Do you mean the subjective feeling of economic and financial insecurity?

Answer:Because personal income is different, economic accessibility refers to subjectively perceived accessibility by the elderly.

12. Most literature use Dummies for modeling the different educational levels . Please define them and report estimates, specifying the omitted one. The same for residence, which is the omitted one? It seems Urban but again, please specify the omitted in the variable definition section.

Answer:All variables are ordered variables, measured by Likert's five-point method.

we add a list of options in Table 2.

13. Row 256 : "However, health status and gender have no significant impact on the realized utilization, indicating that the impact of health status and gender on the realized utilization has not yet shown." Not clear. Please explain the last sentence.

Answer:Health status and gender have no significant impact on the realized utilization, indicating that the differences in gender and health status are non significant after controlling for other socioeconomic determinants of utilization.

14. Please provide a definition of the health status variable and how the Likert scale is derived.Is it self-assessed health, or the n. of chronic conditions or objective health status?

Answer:it is self-assessed health.

15. Results in Table 3 seem to come from a simple logistic model not from a multinomial logit.

Answer:It is ordered logit.

16. The effect of the interactions of the family support variable with all determinants as a final model including all interactions need to be added for completeness.

Answer:The adjustment variables studied in this article are adjustments to economic accessibility, not all variables.

17. I would suggest to not go beyond the results obtained and cut/ heavily reduce this part discussion. 

Answer:Because high-income older adults reduce their demand for basic healthcare services, this article proposes relevant suggestions to increase the utilization of healthcare services by high-income elderly people.

18. There are here a series of value judgemental considerations that are not linked to the results that cannot be supported by the research done.

Answer: Based on the research results, this article proposes corresponding policy recommendations from the perspectives of reducing the cost of healthcare services, improving the economic accessibility of people with different incomes, and increasing family support.

19. "Finally, our aims are to transform potential needs into realized consumption behaviors, enlarge consumption market and upgrade consumption structure.(row 322)" is not linked with the paper aims and results.

Answer: Change upgrade consumption structure to increase utilization rate.

20. I think this cannot be done with the elderly if they are sick and frail. No insurance will sell packages to the over 65.

Answer: For the elderly in good health, delaying retirement is one of the effective ways to increase income, and it can improve the mental health of the elderly.

thanks for your suggestions!

Reviewer 3 Report

General comments

=============

The paper “The Inverted U-shaped Effect of Economic Accessibility on Realized Utilization of Home-Based Healthcare Services for the Older Adults in China: the Moderating Role of Family Support” aims to put forward „the inverted U-shaped hypothesis about the realized utilization and economic accessibility of healthcare services and introduces the moderating variable of family support.“ (see p. 2, line 49-51)

First of all, I would propose to change the title and to rephrase the title to make it shorter. ”The Impact of Economic Accessibility on Realized Utilization of Home-Based Healthcare Services for the Elderly in China: an Inverted U-shaped Relationship” or something of the sort.

It is not clear how the respondents were recruited and in which form the survey took place. Data collection was done through a so-called “summer survey of major national projects in 2019” (see p. 4). A sample of n=832 questionnaires was left for analysis. In order to analyze the hypothesized inverted u-shaped function between economic accessibility and realized utilization of home-based healthcare services, multiple logit regression models were calculated.

Unfortunately, the exact wording of the applied questionnaires was not supplied and, as a consequence, the methods are difficult to understand and not completely comprehensible.

With regard to the applied methodology, I am not well versed in this specific topic and unfortunately I am not a statistician. From an outside perspective, however, I would question the methodological testing of the hypothesized inverted u-shaped functions and would also urge the authors to make the manuscript more concise and more scientifically sound. It partially reveals a golden thread, but some sentences are very difficult to understand. Especially also the reasoning for the hypothesized inverted u-shaped function of the proposed relationship between economic accessibility and realized utilization of home-based healthcare services is not comprehensible. The proposed relationship, however, constitutes the centerpiece of the manuscript under review.

If the authors manage to address the raised concerns properly, however, then the present manuscript could be an interesting contribution to the target journals of Healthcare.  

To sum up, in relation to the present manuscript, there are some major weaknesses, which have to be resolved before it can be seriously considered for publication in Healthcare MDPI. The manuscript in general is reasonably well written with regard to style and grammar, and therefore I would like to elaborate on the weaknesses of the present manuscript in detail as follows and invite the authors to rework it in a thorough revision round.

My comments are described in detail below.

Specific comments

=============

Major and minor comments

---------------------

Please regard the following points as constructive criticism.

 Originality/Novelty: 

  1. With regard to its novelty/originality, the study proposes an inverted u-shaped relationship between economic accessibility and realized utilization of home-based healthcare services. This might be an interesting question, but I do not understand its rationality and the derivation of the respective hypothesis (H1). This may be caused by the fact that the study totally lacks a theoretical foundation. Although the Anderson model is mentioned as having been used for other empirical studies in the field, no alternative theoretical framework is brought up in the present manuscript. Thus, the whole empirical study seems to me more or less data driven.
  2. At the beginning, I would propose to add some actual figures on the aging population in China (see e.g., Fu et al., 2018), to underline the importance of the present topic. The paper by Fu et al. (2018) is very interesting because it also uses the cut-off-point of 60 years for the definition of “elderly” and thus delivers support for this threshold used in the present manuscript under consideration. Bearing in mind that overaging is a problem in many countries all over the world and that the present pandemic situation additionally jeopardizes the vulnerable target group of elderly, the focus of the empirical study is up-to-date. In this respect, the study provides an advance in the field.

Significance:

  1.  In relation to the manuscript’s significance, the results are interpreted appropriately. However, I have to question the applied methodology to identify an inverted-u-shaped function. There is some literature with regard to theorizing and testing inverted u-shaped relationships, like e.g. Haans et al. (2016) or Simonsohn (2018, 2019). I would cordially invite the authors to read these papers carefully and look especially at Table 4 (Checklist for theorizing and testing moderation with inverted U) in the paper by Haans et al. (2013). A recently published Healthcare-paper by Liu et al. (2020), which had a completely different research focus, could be interesting for the authors as the authors also tested an inverted u-shaped function in the area of health-services. The authors are cordially invited to look at the depiction and visualization of the conceptual model and how the hypotheses were drawn out of literature in this published paper.
  2. I miss a clear explanation of the interesting constructs in the manuscript under review. The authors should elaborate on this.
  3. The reasoning of the proposed inverted u-shaped function should be developed and argued thoroughly and based on existing literature, at least to some extent. This would enhance the readability and comprehensibility of the manuscript under consideration.

Quality of Presentation:

  1. With regard to the quality of presentation, the article is at least partially written in an appropriate way, but some of the sentences are not comprehensible.
  2. I would appreciate a visualization of the hypotheses in Figure 1 with a list of indicators for each construct included in the Figure. It should be stated that the independent variable is economic accessibility with the related indicators of elderly income, savings, healthcare costs etc.
  3. With regard to the description of the methodology, I would appreciate a Table with all of the items used and their sources and answer options. There are some inconsistencies between the text and Table 2. Economic accessibility, e.g., seems to have been measured with four items. Was the sum score then used for the further procedures? What was the answer option for these items? From 1=weak to 5=strong does not make any sense. Thus, I would amend Table 2 and include some additional information with regard to the operationalization of each construct.
  4. In Table 2, I would recommend inserting the frequencies of the dependent variable (DV) service utilization to the mean value. Thus, I would recommend to insert a new column and to report mean (SD) and n (%) as two separate rows and report both for the DV and IV and the control variables.
  5. With regard to the calculations, I think it would be a fruitful endeavor to distinguish users and nonusers from each other. How many users and nonusers are there with regard to realized utilization? How was it measured?
  6. Cronbach’s alpha does not reflect validity, but reliability (see p. 4). I do not understand the reasoning for a strengthening effect of family support. One reason could be that the average reader might not know how families in China are constituted. With regard to its measurement, family support seems to be reflected by the number of people in one’s family. I think, much more information should be given in the Introduction about the situation in China (overaging population, healthcare system and how it gets financed, family situation in China, situation of elderly in China with regard to their sociodemographic and psychographic background). Maybe also some facts about COVID-19 could be mentioned, because access to home-based healthcare could be crucial in pandemic times. The authors should elaborate on this.
  7. The authors should argue why they used multiple logit regression models for calculation purposes and not the ones used by Liu et al. (2020).

Scientific Soundness:

  1. With regard to its scientific soundness, it is not comprehensible to me how the Respondents were recruited. The data collection is the result of a commendable effort, but it is not well explained how the data collection took place. Did the authors deliver any incentives? How many respondents were excluded from analysis due to which reasons? How was the data check executed?
  2. The Literature review should reveal a better golden thread and the subheadings should not be overlapping. Is there a theoretical background for the influencing chain, from which the hypotheses could be drawn out?
  3. The Results section is a mixture of revealing results and interpreting them at the same time. I would propose to be consistent and report the results in the Results section and afterwards interpret and discuss them in the Discussion section, but not to mix both in the Results section. Afterwards, a Conclusion and a Limitations section should follow. In the present form of the manuscript, all of the sections are intermingled and thus, the structure is really confusing.
  4. In the Discussion section you should definitely stick to the results and try not to be subjective and evaluative.
  5. Additionally, as has been mentioned above, a Limitations section should be added.

Interest to the Readers:

With respect to the interest of the readers, the conclusions in general as well as the practical implications could be fruitful and interesting. I hope that the authors will manage to visualize and back up their hypotheses more comprehensibly and find a proper method for identifying a u-shaped function of relationship in a thorough revision round. 

Overall Merit:

In terms of overall merit, I think that in the event that the authors manage to tackle the problems raised (like, e.g., inappropriate reasoning and testing of an inverted u-shaped relationship), the manuscript under consideration could make a valuable contribution to the research field.

English Level:

I am not a native speaker, but there are several  grammatical flaws in the manuscript as far as I can judge from a non-native speaker perspective.

To sum up, the theoretical background should cohere with the results and clear research questions should run like a golden thread through the whole manuscript. I see a lot of work requiring a diligent revision round, but I hope that the authors will address all of the raised concerns properly to make this interesting study publishable in Healthcare in the near future.

Good luck with your research!

References:

Haans, R. F., Pieters, C., & He, Z. L. (2016). Thinking about U: Theorizing and testing U‐and inverted U‐shaped relationships in strategy research. Strategic Management Journal, 37(7), 1177-1195.

Liu, J., Zhang, X., Kong, J., & Wu, L. (2020). The Impact of Teammates’ Online Reputations on Physicians’ Online Appointment Numbers: A Social Interdependency Perspective. In Healthcare (Vol. 8, No. 4, p. 509). Multidisciplinary Digital Publishing Institute.

Simonsohn, U. (2018). Two lines: A valid alternative to the invalid testing of U-shaped relationships with quadratic regressions. Advances in Methods and Practices in Psychological Science, 1(4), 538-555.

Simonsohn, U. (2019). "Two lines: A valid alternative to the invalid testing of U-shaped relationships with quadratic regressions": Corrigendum. Advances in Methods and Practices in Psychological Science, 2(4), 410–411.

Author Response

1. First of all, I would propose to change the title and to rephrase the title to make it shorter. ”The Impact of Economic Accessibility on Realized Utilization of Home-Based Healthcare Services for the Elderly in China: an Inverted U-shaped Relationship” or something of the sort.

Answer: rephrase the title to “Impact of Economic Accessibility on Realized Utilization of Home-Based Healthcare Services for the Older Adults in China.”

2. It is not clear how the respondents were recruited and in which form the survey took place.

Answer: The status quo was investigated by stratified sampling. These members have professional backgrounds in elderly care services and have certain investigative capabilities.

3. I do not understand its rationality and the derivation of the respective hypothesis (H1). This may be caused by the fact that the study totally lacks a theoretical foundation.

Answer: Based on Anderson model, this article takes the research results of related scholars as the breakthrough point, discovers the inverted U-shaped relationship between economic accessibility and the utilization of healthcare services, and then proposes hypotheses.

4. At the beginning, I would propose to add some actual figures on the aging population in China, to underline the importance of the present topic.

Answer: By the end of 2019, the Chinese population of 60 years old and above was close to 254 million people, accounting for 18.1% of the total population of China. China is facing a series of aging problem, such as labor shortage, insufficient pension and so on.

5.What was the answer option for these items? From 1=weak to 5=strong does not make any sense.

Answer:  Add a list of options.

6. With regard to its scientific soundness, it is not comprehensible to me how the Respondents were recruited. The data collection is the result of a commendable effort, but it is not well explained how the data collection took place.

Answer: The status quo was investigated by stratified sampling, and investigate three representative regions of Hanzhong, Baoji, and Yan'an. These members have professional backgrounds in healthcare services and have certain investigative capabilities. Considering the integrity and accuracy of data, we obtain 832 valid questionnaires.

7. I would propose to be consistent and report the results in the Results section and afterwards interpret and discuss them in the Discussion section, but not to mix both in the Results section.

Answer: 4.1 and 4.2 are results section, and 4.3 is discussion section.

8. Additionally, as has been mentioned above, a Limitations section should be added.

Answer:  Increase limitations section.

9. In the Discussion section you should definitely stick to the results and try not to be subjective and evaluative.

Answer: Based on the research results, this article proposes corresponding policy recommendations from the perspectives of reducing the cost of healthcare services, improving the economic accessibility of people with different incomes, and increasing family support.

thanks for your suggestions!

Round 2

Reviewer 2 Report

Review Healthcare MDPI ms#1067841  n.2

Due date: 29th January 2021

Dear Authors,

Thank you for submitting a revised version of the manuscript with the former title " “The paper “The Inverted U-shaped Effect of Economic Accessibility on Realized Utilization of Home-Based Healthcare Services for the Older Adults in China: the Moderating Role of Family Support”, modified to “Impact of Economic Accessibility o Realized Utilization of Home-Based Healthcare Services for the Older Adults in China”.

I went through your responses to the comments that I wrote for the first review round. The manuscript lacks to fully address the reply to all the issues raised in the first review round.

Unfortunately, you have addressed only partly to my comments, while not fully addressing  other relevant issues at the same time. This also holds for the other referee report, where you only replied to 9 comments out of 23 made in the first revision round.

Some replies appear not precise and complete sentences are lacking.

I think that the manuscript still needs to be improved with major revisions.

My opinion is that you should revise further the paper and carefully reply and deal with the main points of criticism raised in the first revision round in order for your Manuscript to be taken into consideration for publication in Healthcare MDPI.

Major and minor remarks hereinafter.

Major remarks

Reply to your answers hereinafter:

  1. Row 41 and other parts of the manuscript: The Authors use the term "realized utilization", it is not crear the definition of what you mean by this, I wonder if you mean " observed utilization of public community healthcare services" in practice, but, as this is a dependent variable in the estimated models, a theoretical definition / references are needed in the text e.g. at the beginning of par. 3.2., before the description of how it is measured (row n. 167).

Answer:The realized utilization of healthcare services refers to the use of public facilities and care services of home-based healthcare services for the older adults.

Reply : Still a theoretical definition is missing as it has been raised also by the other referee report.

  1. "Family support strengthens the inverted U-shaped effect of the economic accessibility on realized accessibility of healthcare services for the older adults."
    I would add: while increasing inequalities in access and use.

Answer:Family support not only strengthens the rise in the use of utilization when economic accessibility is insufficient, but also accelerates the decline in the use of healthcare services when economic access is sufficient.

Reply : I think that, especially among the poorer, inequalities between those who have and those who have not family support should be considered.

  1. There appear to be a trade-off here between efficiency ( shifting the burden of caring for the elderly to their family can be cost saving for the government) and equity (looking at those who do have a family support the economic accessibility gap is higher between the poorer and the richer than considering those who have low family support) (Figure 1) 
    Thereby, increasingly having people counting on their family would a) reduce public expenditure (increase in efficiency) but 2) increase socioeconomic inequalities in utilization (with a horizontal equity in utilization reduction). The equity implications need to be discussed in the discussion section.

Answer: Although fairness is important factor affecting service utilization, it is not the focus of this article. Fairness is the focus of subsequent research.

Reply: I understand, nevertheless as you derive policy implications and indications it would be important to stress that there can be equity implications and to mention further work to be done  in the limitations section.

  1. The Authors use the term "economic accessibility of healthcare services". I would suggest to use the term "affordability" which seems more appropriate.

Answer:Affordability emphasizes personal income, while economic accessibility includes both personal income and the cost of services.

Reply: Please refer to definitions from the literature in the text. For example from the WHO: “Economic accessibility, or affordability “is a measure of people’s ability to pay for services without financial hardship. It takes into account not only the price of the health services but also indirect and opportunity costs (e.g. the costs of transportation to and from facilities and of taking time away from work).” Affordability is influenced by the wider health financing system and by household income.” (https://www.who.int/gender-equity-rights/understanding/accessibility-definition/en/).

See also UNITED STATES OF CARE, WHAT IS “AFFORDABLE” HEALTH CARE? A review of concepts to guide policymakers, U.Penn, https://ldi.upenn.edu/sites/default/files/pdf/Penn%20LDI%20and%20USofC%20Affordability%20Issue%20Brief_Final.pdf , accessed Jan 29th 2020.

  1. "is inverted U-shaped, not a continuous promotion." Not clear what the Authors mean (maybe substitute "promotion" with "one"?)

Answer:Change continuous promotion to linear positive effect.

Reply: this is fine for me, thank you.

  1. "exerting the role of family". Not clear what the Authors mean (e.g. supporting families with public policies? incentives to family to take care of the elderly?)

Answer:The role of family refers to the healthcare services for the elderly provided by the family.

  1. "spiritual consolation" or "spiritual comfort". Not clear what the Authors mean. Please explain more in detail.

Answer:Spiritual consolation refers to the mental needs and spiritual entertainment of the elderly.

  1. Row 108 "the types of healthcare services are single" Not clear what the Authors mean. Please explain more in detail.

Answer:Change types to contents.

Reply to answers 6-8: please further clarify. The English style has not significantly been improved

  1. "realized utilization of healthcare services" Again, not clear what the Authors mean, maybe "utilization of public healthcare services"?

Answer:Change healthcare services to public healthcare services.

Reply: thank you this is fine for me.

  1. "To meet the needs of diverse healthcare services" Not clear what it is meant by "diverse" and the overall text.

Answer:Increase the contents of special nursing, rehabilitation nursing and so on to explain the diverse healthcare services.

Reply: the English style still needs to be improved

  1. Not clear what the Authors mean by "older adults' subjective perception of economic accessibility". Do you mean the subjective feeling of economic and financial insecurity?

Answer:Because personal income is different, economic accessibility refers to subjectively perceived accessibility by the elderly.

Reply: here you refer only to income and not to costs. The  definition of economic accessibility/affordability is not clear, see also my reply n.4 above.

  1. Most literature use Dummies for modeling the different educational levels . Please define them and report estimates, specifying the omitted one. The same for residence, which is the omitted one? It seems Urban but again, please specify the omitted in the variable definition section.

Answer:All variables are ordered variables, measured by Likert's five-point method.

we add a list of options in Table 2.

Reply: table 2 now is clearer thank you.

However, it is not clear how the independent variables are used. Some should be used as dummy variables not as continuous variables (e.g. urban/rural; male/female ). I think for clarity a full set of estimation results should be reported in an appendix/supplementary material.

  1. Row 256 : "However, health status and gender have no significant impact on the realized utilization, indicating that the impact of health status and gender on the realized utilization has not yet shown." Not clear. Please explain the last sentence.

Answer:Health status and gender have no significant impact on the realized utilization, indicating that the differences in gender and health status are non significant after controlling for other socioeconomic determinants of utilization.

Reply: this is clearer, thank you.

  1. Please provide a definition of the health status variable and how the Likert scale is derived.Is it self-assessed health, or the n. of chronic conditions or objective health status?

Answer:it is self-assessed health.

Reply: : this is clearer, thank you.

  1. Results in Table 3 seem to come from a simple logistic model not from a multinomial logit (you seem to have more than one outcome if I understood clearly). Can you please explain more the choice of the modelling strategy?
    Overall, if I understood that you used a simple logistic model for the probability of outcome , it seems more appropriate to use ordered logit models as you seem to have an ordered dependent variable..

Answer: It is ordered logit.

Reply: I think for clarity a full set of estimation results should be reported in an appendix/supplementary material.Please then include all relevant parameters estimates (the threshold / cut off parameters of the ordered logits and the constant term ).

Please include a complete formula for the ordered logit models. This is for example a paper on a different topic but where ordered logits estimates are clearly presented ( see e.g. Bernd and Hiroyuki, 2011)

  1. The effect of the interactions of the family support variable with all determinants as a final model including all interactions need to be added for completeness.

Answer: The adjustment variables studied in this article are adjustments to economic accessibility, not all variables.

Reply: I disagree. It is important to control for the interaction of the other determinants with family support to see if the interaction between economic accessibility and family comp. remains significant once controlled for the above.

17.Rows 290-324
I would suggest to not go beyond the results obtained and cut/ heavily reduce this part discussion. 

Text omitted by Authors: “For example: row 292 : "The supply of healthcare services should pay attention to diversification and high quality. The consumption market for healthcare services should be opened through the development of door-to-door services or professional healthcare services. This suggestion is applicable to developing countries where healthcare services for the older adults are at an early stage" 
This has no link with the role of family as a mediator in healthcare utilization and the study do not give any evidence from the supply side.

For example, from row 311: 
"From the perspective of the demand side of healthcare services for the older adults, in order to improve the economic accessibility of healthcare services, we should also encourage the older adults to re-employed, invest financial project, buy commercial insurance, and delay retirement, in addition to increasing the coverage and protection of basic old-age insurance and medical insurance. These actions are beneficial to realize the simultaneous growth of the income of the older adults and economic development. For low- income older adults, intergenerational communication, social networks, and media pro motion can be used to change the consumption awareness and consumption habits of the older adults, which will reduce the perceived burden of the older adults, and improve the realized utilization of healthcare services. For high-income older adults, we should remove the information barriers for the older adults to use healthcare services, and make it easier for the older adults to obtain safe and accurate consumption information. " “

Answer: Because high-income older adults reduce their demand for basic healthcare services, this article proposes relevant suggestions to increase the utilization of healthcare services by high-income elderly people.

Reply: your answer is incomplete you only took a part of the text and suggestions were not considered.

  1. There are here a series of value judgemental considerations that are not linked to the results that cannot be supported by the research done. Text omitted: “I do not see why only for high income and not for the poorer it is important to "...make it easier for the older adults to obtain safe and accurate consumption information."
    and the following statement:"Finally, our aims are to transform potential needs into realized consumption behaviors, enlarge consumption market and upgrade consumption structure.(row 322)" is not linked with the paper aims and results.)”

Answer: Based on the research results, this article proposes corresponding policy recommendations from the perspectives of reducing the cost of healthcare services, improving the economic accessibility of people with different incomes, and increasing family support.

Reply: your answer is quite incomplete and suggestions given not considered.

  1. "Finally, our aims are to transform potential needs into realized consumption behaviors, enlarge consumption market and upgrade consumption structure.(row 322)" is not linked with the paper aims and results.

Answer: Change upgrade consumption structure to increase utilization rate.

Reply: the modification of the text you propose  does not  reply to my point

The following text was omitted:

Row: 311: "From the perspective of the demand side of healthcare services for the older adults, in order to improve the economic accessibility of healthcare services, we should also encourage the older adults to re-employed, invest financial project, buy commercial insurance, and delay retirement, in addition to increasing the coverage and protection of basic  old-age insurance and medical insurance."
And :
Row n. 352: "Government should encourage the older adults to increase the income and reduce the economic barriers."

  1. I think this cannot be done with the elderly if they are sick and frail. No insurance will sell packages to the over 65

Answer: For the elderly in good health, delaying retirement is one of the effective ways to increase income, and it can improve the mental health of the elderly.

Reply: Again, these statements are not supported  by your results.

Finally, you did not  reply neither included in the text these suggestions in the first review:

  1. Row 211- "In terms of realized utilization, the mean and median are 2.99 and 3 respectively, indicating that the realized utilization of healthcare services is average and the consumption market for healthcare services has not yet formed." Please explain.
  2. Table 3- please substitute the variable names in the first column for better clarity
  3. Most literature use dummies for modelling the different educational levels . Please define them and report estimates, specifying the omitted one. The same for residence, which is the omitted one? It seems Urban but again, please specify the omitted in the variable definition section.
  4. Row 256 : "However, health status and gender have no significant impact on the realized utilization, indicating that the impact of health status and gender on the realized utilization has not yet shown." Not clear. Please explain the last sentence. Maybe the system is equitable and differences in gender and health status are non significant
  5. 4-DISCUSSION

Row 290 "For high-income elderly groups, healthcare 290 costs are no longer the main reason affecting realized utilization of healthcare services." This is an expected and well known results in the empirical literature on the demand for health and healthcare services, so I would say that this confirms existing literature findings.

In Europe Welfare systems allow the elderly to retire and to choose if they want to continue to work or not, because it is well known that retirement is a right and that it can be necessary for several reasons like, e.g., due to worsening healthcare conditions and not only for this reason. Seeing elderly sick people working to keep up income is not a great society achievement. 
So, I would strongly suggest to make the statements less general and more specific, e.g., giving incentives to the healthy retired elderly population on a voluntary - not compulsory basis- to go back to work for periods/part-time etc..

Moreover, I agree with the other referee main remarks not covered above.

Minor remarks

Row 202 : realizedutilization —>realized utilization

Row 259: The impact of residence on realized utilization is significant at 0.2537,  —> correct with : is 0.2537 and it is significant.

Row 5.2. Limatations  —> limitations

Best regards.

References

  • Bernd Hayo, Hiroyuki Ono, Livelihood and care of the elderly: Determinants of public attitudes in Japan,Journal of the Japanese and International Economies, Volume 25, Issue 1, 2011,Pages 76-98,ISSN 0889-1583, https://doi.org/10.1016/j.jjie.2010.11.001.
  • UNITED STATES OF CARE, WHAT IS “AFFORDABLE” HEALTH CARE? A review of concepts to guide policymakers, U.Penn, https://ldi.upenn.edu/sites/default/files/pdf/Penn%20LDI%20and%20USofC%20Affordability%20Issue%20Brief_Final.pdf , last access Jan 29th
  • Universal Health Coverage and Universal Access, Bulletin of the World Health Organization 2013; 91:546-546A

Author Response

thankyou for your suggestion, sincerely.

  1. The Authors use the term "realized utilization", it is not clear the definition of what you mean by this, I wonder if you mean " observed utilization of public community healthcare services" in practice, but, as this is a dependent variable in the estimated models, a theoretical definition / references are needed in the text e.g. at the beginning of par. 3.2., before the description of how it is measured.

Answer:Referring to the definition of Andersen and Babette [6-7], the realized utilization of healthcare services refers to the use of public facilities and care services of home-based healthcare services for the older adults.

  1. "Family support strengthens the inverted U-shaped effect of the economic accessibility on realized accessibility of healthcare services for the older adults." I think that, especially among the poorer, inequalities between those who have and those who have not family support should be considered.

Answer:Family support not only strengthens the rise in the use of utilization when economic accessibility is insufficient, but also accelerates the decline in the use of healthcare services when economic access is sufficient.

In 2.4, increase “Especially among the poorer, inequalities between those who have and those who have not family support increase differences in realized utilization of healthcare services.”

In 4.3, increase some discussions about efficiency and equity.

  1. Nevertheless as you derive policy implications and indications it would be important to stress that there can be equity implications and to mention further work to be done in the limitations section.

Answer: In 4.3, increase some discussions about efficiency and equity.

In limitations section, increase “evaluation of efficiency and equity for resource utilization”.

  1. The Authors use the term "economic accessibility of healthcare services". I would suggest to use the term "affordability" which seems more appropriate.

Answer: In 2.2, according to the suggestion of reviewer, increase the definition of economic accessibility, “Based on the definition of the World Health Organization, economic accessibility is a measure of people’s ability to pay for services without financial hardship, and takes into account not only the price of the health services but also indirect and opportunity costs.”

  1. Row 108 "the types of healthcare services are single" Not clear what the Authors mean. Please explain more in detail.

Answer:Change "the types of healthcare services are single" to “the contents of healthcare services mostly focus on daily care, which are single and the quality is low.”

  1. "To meet the needs of diverse healthcare services" Not clear what it is meant by "diverse" and the overall text.

Answer:Change "To meet the needs of diverse healthcare services" to “To meet the needs of special nursing, rehabilitation nursing and other healthcare services,”

  1. Not clear what the Authors mean by "older adults' subjective perception of economic accessibility". Do you mean the subjective feeling of economic and financial insecurity?

Answer:In 2.2, increase the definition of economic accessibility.

Family support can supply different care services, so it can reduce dependence on the economy, thereby, it can influence the older adults' subjective perception of economic accessibility. For example, to the poor elderly, if family support can totally meet his needs to healthcare services, he doesn’t need to consider his economic accessibility.

  1. it is not clear how the independent variables are used. Some should be used as dummy variables not as continuous variables (e.g. urban/rural; male/female). I think for clarity a full set of estimation results should be reported in an appendix/supplementary material.

Answer: It is ordered logit. Increase a series of estimation results (LR / cut off parameters of the ordered logit and the constant term ) in table 2 and table 3.

Among them, for the two categorical variables of residence and gender, we set as dummy variables with male=1, female= 0, and urban= 1, rural=0.

  1. Results in Table 3 seem to come from a simple logistic model not from a multinomial logit (you seem to have more than one outcome if I understood clearly). Can you please explain more the choice of the modelling strategy? Overall, if I understood that you used a simple logistic model for the probability of outcome , it seems more appropriate to use ordered logit models as you seem to have an ordered dependent variable.

Answer: It is ordered logit.

In Model 1-6, the control variables are successively added to control the stability of the model results.

In Model 7-12, we test the impact of the cross item of family support and squared economic accessibility on realized utilization when the control variables are successively added to control the stability of the model results.

In Model 13, under the interactions of the family support variable with all determinants, the impact of the cross item of family support and squared economic accessibility on realized utilization is 0.0593.

  1. Please include a complete formula for the ordered logit models. This is for example a paper on a different topic but where ordered logits estimates are clearly presented ( see e.g. Bernd and Hiroyuki, 2011)

Answer: In 4, increase a complete formula for the ordered logit models.

  1. The effect of the interactions of the family support variable with all determinants as a final model including all interactions need to be added for completeness.

Answer: Increase the effect of the interactions of the family support variable with all determinants as Model 13. In Model 13, under the interactions of the family support variable with all determinants, the impact of the cross item of family support and squared economic accessibility on realized utilization is 0.0593.

  1. I would suggest to not go beyond the results obtained and cut/ heavily reduce this part discussion. There are here a series of value judgemental considerations that are not linked to the results that cannot be supported by the research done.

Answer: In 4.3,the discussion are divided into two parts of the impact of economic accessibility on realized utilization and the moderating role of family support. The first part proposes improvement suggestions from three perspectives of service subject responsibility, economic accessibility of low-income groups and economic accessibility of high-income groups. The second part analyzes the role of family support from the perspectives of efficiency and equity, and puts forward some suggestions.

  1. "In terms of realized utilization, the mean and median are 2.99 and 3 respectively, indicating that the realized utilization of healthcare services is average and the consumption market for healthcare services has not yet formed." Please explain.

Answer: In terms of realized utilization, the mean and median are 2.99 and 3 respectively, indicating that the frequency of realized utilization of healthcare services has not reached a high level and the older adults' awareness of using community healthcare services has not yet formed.

  1. Table 3- please substitute the variable names in the first column for better clarity.

Answer: In Table 2 and Table 3, substitute the variable names in the first column

  1. Most literature use dummies for modelling the different educational levels . Please define them and report estimates, specifying the omitted one. The same for residence, which is the omitted one? It seems Urban but again, please specify the omitted in the variable definition section.

Answer: Educational level is divided into five parts of elementary school and below, junior high school, high school, college and bachelor degree and above. So it belongs to ordered variable. For the two categorical variables of residence and gender, we set as dummy variables with male=1, female= 0, and urban= 1, rural=0.

  1. Row 256 : "However, health status and gender have no significant impact on the realized utilization, indicating that the impact of health status and gender on the realized utilization has not yet shown." Not clear. Please explain the last sentence. Maybe the system is equitable and differences in gender and health status are non significant.

Answer: health status and gender have no significant impact on the realized utilization, indicating that the differences in gender and health status are not significant after controlling for other socioeconomic determinants of utilization.

  1. In Europe Welfare systems allow the elderly to retire and to choose if they want to continue to work or not, because it is well known that retirement is a right and that it can be necessary for several reasons like, e.g., due to worsening healthcare conditions and not only for this reason. Seeing elderly sick people working to keep up income is not a great society achievement. 

Answer: In 4.3,delete the contents of re-employment of the elderly.

Reviewer 3 Report

Dear authors!

Thank you for submitting a revised version of the manuscript with the former title " “The paper “The Inverted U-shaped Effect of Economic Accessibility on Realized Utilization of Home-Based Healthcare Services for the Older Adults in China: the Moderating Role of Family Support”, which was amended to “Impact of Economic Accessibility o Realized Utilization of Home-Based Healthcare Services for the Older Adults in China”. I have read thoroughly through your responses to my comments in the first review round. Unfortunately, you have addressed only a small proportion of my comments, but you have neglected many others at the same time. To sum up, I think that the manuscript has improved only marginally through the revision and I am not sure whether it could make a nice contribution to literature in the long run. I did not get the impression, that you have invested the appropriate amount of effort for addressing all of the raised issues in the first review round. You only selected 9 comments out of 23 comments made in the first revision round.

 I even do not see whether you responded to the issues raised by the other reviewer. You even partially did not have enough time to use whole sentences in order to respond to the issues raised as points of criticism in the first review round. Thus, I miss, a thorough engagement in the main points of criticism raised in the first revision round. So please look through the concerns still remaining, which should be smoothed out before the manuscript can seriously be taken into consideration for publication in Healthcare MDPI, at least from my point of view.

Please find my comments in detail below.

=============

Major comments

---------------------

  1. The following point of criticism still remains: With regard to its novelty/originality, the study proposes an inverted u-shaped relationship between economic accessibility and realized utilization of home-based healthcare services. This might be an interesting question, but I do not understand its rationality and the derivation of the respective hypothesis (H1). This may be caused by the fact that the study totally lacks a theoretical foundation. Although the Anderson model is mentioned as having been used for other empirical studies in the field, but no alternative theoretical framework is brought up in the present manuscript. Thus, the whole empirical study seems to me more or less data driven.
  2. The following point of criticism still remains: In relation to the manuscript’s significance, the results are interpreted appropriately. However, I have to question the applied methodology to identify an inverted-u-shaped function. There is some literature with regard to theorizing and testing inverted u-shaped relationships, like e.g. Haans et al. (2016) or Simonsohn (2018, 2019). I would cordially invite the authors to read these papers carefully and look especially at Table 4 (Checklist for theorizing and testing moderation with inverted U in the paper by Haans et al. (2013). A recently published Healthcare-paper by Liu et al. (2020), which had a completely different research focus, could be interesting for the authors as the authors also tested an inverted u-shaped function in the area of health-services. The authors are cordially invited to look at the depiction and visualization of the conceptual model and how the hypotheses were drawn out of literature in this published paper. What is also interesting there is a clear explanation of all of the interesting constructs. I miss a clear explanation of the interesting constructs in the manuscript under review. You should elaborate on this.
  3. The following point of criticism still remains: The reasoning of the proposed inverted u-shaped function should be developed and argued thoroughly and based on existing literature, at least to some extent. This would enhance the readability and comprehensibility of the manuscript under consideration.
  4. The following point of criticism still remains: With regard to the quality of presentation, the article is at least partially written in an appropriate way, but some of the sentences are not comprehensible.
  5. The following point of criticism still remains: I would appreciate a visualisation of the hypotheses in Figure 1 with a list of indicators for each construct being also depicted in the Figure. It should be stated, that the independent variable is economic accessibility with the related indicators of elderly income, savings, healthcare costs etc.
  6. The following point of criticism still remains: Economic accessibility, e.g., seems to have been measured with four items. Was the sum score then used for the further procedures? There are several mistakes in Table 2 with regard to the answer options. E.g., health status was measured with a rating scale ranging from 1=very good to 5=very good. Psychosocial status was measured with a scale ranging from 1=very bad, 2=bad, 3=not bad, 4=good, 5=very good. 3 has to be neither good, nor bad. Otherwise the scale would be imbalanced.
  7. The following point of criticism still remains: In Table 2, I would recommend to insert the frequencies of the dependent variable (DV) service utilization to the mean value. Thus, I would recommend to insert a new column and to report mean (SD) and n (%) as two separate rows and report both for the DV and IV and the control variables.
  8. The following point of criticism still remains: With regard to the calculations, I think it would be a fruitful endeavor to distinguish users and nonusers from each other. How many users and nonusers are there with regard to realized utilization? How was it measured?
  9. The following point of criticism still remains: Cronbach’s alpha does not reflect validity, but reliability (see p. 4). I do not understand the reasoning for a strengthening effect of family support. Maybe one reason is, that the average reader might not know how families in China are constituted. With regard to its measurement, family support seems to be reflected by the number of people in one’s family. I think, in the Introduction much more information should be given about the situation in China (overaging population, healthcare system and how it gets financed, family situation in China, situation of elderly in China with regard to their sociodemographic and psychographic background). Maybe also some facts about COVID-19 could be mentioned, because access to home-based healthcare could be crucial in pandemic times. You should elaborate on this.
  10. The following point of criticism still remains: You should argue why they used multiple logit regression models for calculation and not the ones used by Liu et al. (2020).
  11. The following point of criticism still remains: With regard to its scientific soundness, it is not comprehensible to me how the Respondents were recruited. The data collection is the result of a commendable effort, but it is not well explained how the data collection took place. Did the authors deliver any incentives? How many respondents were excluded from analysis due to which reasons? How was the data check executed?
  12. The following point of criticism still remains: The Literature review should reveal a better golden thread and the subheadings should not be overlapping. Is there a theoretical background for the influencing chain, where the hypotheses could be drawn out?
  13. The following point of criticism still remains: The Results section is a mixture of revealing results and interpreting them at the same time. I would propose to be consistent and report the results in the Results section and afterwards interpret and discuss them in the Discussion section, but not to mix both in the Results section. Afterwards, a Conclusion and a Limitations section should follow. In the present from of the manuscript, all of the sections are intertwined and thus, the structure is really confusing.
  14. The following point of criticism still remains: I am not a native speaker, but there are several grammatical flaws in the manuscript as far as I can judge from my non-native speaker perspective.

Besides, you did not refer to the four references recommended by me in the first revision round.

References:

Haans, R. F., Pieters, C., & He, Z. L. (2016). Thinking about U: Theorizing and testing U‐and inverted U‐shaped relationships in strategy research. Strategic Management Journal, 37(7), 1177-1195.

Simonsohn, U. (2018). Two lines: A valid alternative to the invalid testing of U-shaped relationships with quadratic regressions. Advances in Methods and Practices in Psychological Science, 1(4), 538-555.

Liu, J., Zhang, X., Kong, J., & Wu, L. (2020). The Impact of Teammates’ Online Reputations on Physicians’ Online Appointment Numbers: A Social Interdependency Perspective. In Healthcare (Vol. 8, No. 4, p. 509). Multidisciplinary Digital Publishing Institute.

Simonsohn, U. (2019). "Two lines: A valid alternative to the invalid testing of U-shaped relationships with quadratic regressions": Corrigendum. Advances in Methods and Practices in Psychological Science, 2(4), 410–411.

Author Response

Thankyou for your suggestion, sincerely.

  1. With regard to its novelty/originality, the study proposes an inverted u-shaped relationship between economic accessibility and realized utilization of home-based healthcare services. This might be an interesting question, but I do not understand its rationality and the derivation of the respective hypothesis (H1). This may be caused by the fact that the study totally lacks a theoretical foundation. Although the Anderson model is mentioned as having been used for other empirical studies in the field, but no alternative theoretical framework is brought up in the present manuscript. Thus, the whole empirical study seems to me more or less data driven.

Answer: in 2.3 and 2.4, increase some theoretical foundation. Through theoretical analysis and literature review, this article finds the research gap, and then proposes hypotheses. And increase a conceptual framework figure.

  1. In relation to the manuscript’s significance, the results are interpreted appropriately. However, I have to question the applied methodology to identify an inverted-u-shaped function. There is some literature with regard to theorizing and testing inverted u-shaped relationships, like e.g. Haans et al. (2016) or Simonsohn (2018, 2019). I would cordially invite the authors to read these papers carefully and look especially at Table 4 (Checklist for theorizing and testing moderation with inverted U in the paper by Haans et al. (2013). A recently published Healthcare-paper by Liu et al. (2020), which had a completely different research focus, could be interesting for the authors as the authors also tested an inverted u-shaped function in the area of health-services. The authors are cordially invited to look at the depiction and visualization of the conceptual model and how the hypotheses were drawn out of literature in this published paper. What is also interesting there is a clear explanation of all of the interesting constructs. I miss a clear explanation of the interesting constructs in the manuscript under review. You should elaborate on this.

Answer: In 2.3, the first paragraph states that economic accessibility can affect realized utilization. The second paragraph proposes that economic accessibility has a positive effect on healthcare utilization based on literature analysis. The third paragraph proposes that economic accessibility has a negative effect for high-income older adults. Therefore, hypothesis 1 is proposed based on two conclusions.

According to Haans et al. (2016), an inverted U-curve may be constructed by interacting two latent linear functions, one positive and one negative in realized utilization.

  1. The reasoning of the proposed inverted u-shaped function should be developed and argued thoroughly and based on existing literature, at least to some extent. This would enhance the readability and comprehensibility of the manuscript under consideration.

Answer: increase “For basic healthcare service, older adults with different affordability have different using willing and realized utilization [20]. Therefore, economic accessibility on realized utilization is not a linear positive effect.”

[20] Habibov, N. What determines healthcare utilization and related out-of-pocket expenditures in Tajikistan? Lessons from a national survey. Int J Public Health, 2009,54, 260–266. DOI: 10.1007/s00038-009-8044-2.

  1. With regard to the quality of presentation, the article is at least partially written in an appropriate way, but some of the sentences are not comprehensible.

Answer: The authors try their best to modify and improve the quality of presentation in the manuscript.

  1. I would appreciate a visualisation of the hypotheses in Figure 1 with a list of indicators for each construct being also depicted in the Figure. It should be stated, that the independent variable is economic accessibility with the related indicators of elderly income, savings, healthcare costs etc.

Answer: In 2.4, increase a conceptual framework figure with a list of indicators.

  1. The following point of criticism still remains: Economic accessibility, e.g., seems to have been measured with four items. Was the sum score then used for the further procedures? There are several mistakes in Table 2 with regard to the answer options. E.g., health status was measured with a rating scale ranging from 1=very good to 5=very good. Psychosocial status was measured with a scale ranging from 1=very bad, 2=bad, 3=not bad, 4=good, 5=very good. 3 has to be neither good, nor bad. Otherwise the scale would be imbalanced.

Answer: Economic accessibility is measured with four items. The total score is the sum of the scores of the four questions, and the total score of 4-6 is recorded as 1, 7-10 is recorded as 2, 11-14 is recorded as 3, and 15-17 is recorded as 4, 18 -20 is recorded as 5.

change “not bad” to “neither good nor bad” in Table 2.   1=very bad

  1. The following point of criticism still remains: In Table 2, I would recommend to insert the frequencies of the dependent variable (DV) service utilization to the mean value. Thus, I would recommend to insert a new column and to report mean (SD) and n (%) as two separate rows and report both for the DV and IV and the control variables.

Answer: Combine Table 1 and Table 2, and add the frequencies and n (%) of the DV and IV in new Table 1.

  1. With regard to the calculations, I think it would be a fruitful endeavor to distinguish users and nonusers from each other. How many users and nonusers are there with regard to realized utilization? How was it measured?

Answer: we adopt the question of "Do you often use the healthcare services provided by the community" to measure the realized utilization of healthcare services. The number of seldom, sometimes, often, usually, and always respectively is 56,243,270,183,80 in Table 1.

  1. The following point of criticism still remains: Cronbach’s alpha does not reflect validity, but reliability (see p. 4). I do not understand the reasoning for a strengthening effect of family support. Maybe one reason is, that the average reader might not know how families in China are constituted. With regard to its measurement, family support seems to be reflected by the number of people in one’s family. I think, in the Introduction much more information should be given about the situation in China (overaging population, healthcare system and how it gets financed, family situation in China, situation of elderly in China with regard to their sociodemographic and psychographic background). Maybe also some facts about COVID-19 could be mentioned, because access to home-based healthcare could be crucial in pandemic times. You should elaborate on this.

Answer: The sample’s reliability value of Cronbach's α is 0.77, and validity value of Kaiser-Meyer-Olkin (KMO) and Bartlett tests is 0.64, which is representative.

In 2.4, increase “In China, it is a traditional virtue that families provide healthcare services for the elderly. However, with the development of industrialization, urbanization, and the normalization of population mobility, the function of family care for the elderly is gradually weakening, and family support is significantly different in the process of using community healthcare services for the elderly.”

Through COVID-19 can affect income, saving and costs of healthcare service, this article only discusses the impact of economic accessibility on realized utilization under normal circumstances, and special circumstances are the limitations of this article.

In 5.2, increase ”In addition, we can also discuss the impact of economic accessibility on realized utilization with the effect of COVID-19.”

  1. You should argue why they used multiple logit regression models for calculation and not the ones used by Liu et al. (2020).

Answer: In 4, The dependent variable is divided into five categories from weak to strong, which belongs to ordered variables, so it is suitable to adopt ordered logit regression models.

  1. With regard to its scientific soundness, it is not comprehensible to me how the Respondents were recruited. The data collection is the result of a commendable effort, but it is not well explained how the data collection took place. Did the authors deliver any incentives? How many respondents were excluded from analysis due to which reasons? How was the data check executed?

Answer: In 3.1, increase “These members have professional backgrounds in healthcare services and have certain investigative capabilities.”

The status quo was investigated by stratified sampling. The first stage of stratification is based on cities, the second stage of stratification is based on counties, the third stage of stratification is based on townships, and the fourth stage of stratification is based on villages. The fifth stage of stratification takes the elderly as a sample.”

Finally, the team obtained a total of 948 valid questionnaires, and we adopt 832 valid questionnaires related to the theme, by discarding missing data and invalid questionnaires.

  1. The Literature review should reveal a better golden thread and the subheadings should not be overlapping. Is there a theoretical background for the influencing chain, where the hypotheses could be drawn out?

Answer: In 2.3, increase Literature review of “From the research results, the above literature believes that economic accessibility has a positive effect on realized utilization. ”  

increase “For basic healthcare service, older adults with different affordability have different using willing and realized utilization. Therefore, economic accessibility on realized utilization is not a linear positive effect.”

In 2.3, the first paragraph states that economic accessibility can affect realized utilization. The second paragraph proposes that economic accessibility has a positive effect on healthcare utilization based on literature analysis. The third paragraph proposes that economic accessibility has a negative effect for high-income older adults. Therefore, hypothesis 1 is proposed based on two conclusions.

  1. The Results section is a mixture of revealing results and interpreting them at the same time. I would propose to be consistent and report the results in the Results section and afterwards interpret and discuss them in the Discussion section, but not to mix both in the Results section. Afterwards, a Conclusion and a Limitations section should follow. In the present from of the manuscript, all of the sections are intertwined and thus, the structure is really confusing.

Answer: In the results section, this article analyzes the empirical results and reasons. In the discussion section, it focuses on policy and suggestions to improve the utilization rate of healthcare services.

In 4.2 and 4.3, The author makes major revision.

  1. I am not a native speaker, but there are several grammatical flaws in the manuscript as far as I can judge from my non-native speaker perspective.

Answer: The authors try their best to modify and improve the grammar flaws in the manuscript.

  1. Besides, you did not refer to the four references recommended by me in the first revision round.

Answer: increase an important reference. According to Haans et al. (2016), an inverted U-curve may be constructed by interacting two latent linear functions, one positive and one negative in realized utilization.

Haans, R. F., Pieters, C., & He, Z. L. (2016). Thinking about U: Theorizing and testing U‐and inverted U‐shaped relationships in strategy research. Strategic Management Journal, 37(7), 1177-1195.DOI: 10.1002/smj.2399.

Round 3

Reviewer 2 Report

I have quickly read the manuscript. I think that now the manuscript has been improved  and they replied to my questions so that it could be published.

Author Response

thanks for your suggestion, again!

Reviewer 3 Report

See the attachment.

Author Response

thanks for your suggestion, again!

  1. The following points of criticism with regard to an incorrect justification of reliability, validity and representativeness still remain: As I have mentioned in the former review round, Cronbach’s alpha does not reflect validity, but reliability (see p. 4). You have corrected the term “validity” and replaced it by the proper term “reliability” instead. In the new version of the sentence, however you amend the main clause with the following subclause: “and validity value of Kaiser-Meyer-Olkin (KMO) and Bartlett tests is 0.64, which is representative.” This subclause is misleading. The KMO and Bartlett test both check the data prerequisites of a consecutive exploratory factor analysis. The value of 0.64 confirms (i.e. in case that the value is above 0.50), that the included variables are appropriate for calculating an exploratory factor analysis in a further step. I see, however, neither the necessity or reasoning for conducting an EFA nor which variables could be useful for calculating an EFA. Why do you report these measures (KMO and Bartlett test)? Additionally, both of the measures neither reflect nor guarantee representativeness. You should cancel the inserted subclause instead.

Answer: Delete the subclause “and validity value of Kaiser-Meyer-Olkin (KMO) and Bartlett tests is 0.64, which is representative.”

the sample’s reliability value of Cronbach's α is 0.77,which confirms that the included variables are appropriate for calculating a further analysis.

  1. I invited you to add a separate column in Table 1 for additionally reporting frequencies and percentages for the measured variables. It does not make sense at all, however, to calculate means for ordinal and nominal scaled variables. Thus, you should cancel the means for gender, education level, residence and also for age (as it was measured in categories and not in absolute terms) and cancel also the respective standard deviation. For all of these ordinal and nominal scaled variables, only the frequencies and percentages should be reported. For all of the others, i.e. the - more or less - interval scaled variables (realized utilization, economic accessibility, family support, psychosocial status, health status), both frequencies/percentages and mean/standard deviation should be reported. The median, however, can be reported for all of the variables independently on which scale level they have been measured.

Answer: according to the suggestion, cancel the means, median and standard deviation for gender, education level, residence and age in Table 1.

  1. In Table 2, the newly inserted lines cut 1, cut 2, cut 3 and cut 4 should be explained. What was the reason for inserting these lines, what does “cut 1”, “cut 2”, “cut3”, and “cut 4” mean and how can the results be interpreted with regard to these new lines? Additionally, the additional Model 13 is not explained properly. What does Model 13 refer to and how can the results be interpreted? Why did you add a new Model 13? What is the reasoning for doing this?

Answer: the cut values of dependent variable are meaningless to explain the impact between two variables, and these values only ensure completeness of the output result.

Model 13 includes the interactions of the family support variable with all variables, and can make sure the result is stable to support Hypothesis 2 and not affected by other variables.

“In Model 13, under the interactions of the family support variable with all control variables, the impact of the cross item of family support and squared economic accessibility on realized utilization is 0.0593.The results are stable with the introduction of different cross item and are significant at 1% level, indicating that the family support has strengthened the inverted U-shaped effect of the economic accessibility on realized utilization of healthcare services. Hypothesis 2 is supported.”

  1. I appreciate that you have defined “economic accessibility” and “realized utilization of healthcare services” in the revised version of the manuscript as I have proposed. Unfortunately, I miss a thorough definition of “psychosocial status”. How was it explained in the questionnaire and what does this construct refer to? Thus, you are cordially invited to define also “psychosocial status” in the next review round.

Answer: increase the definition of “the psychosocial status refers to the attitudes and emotions of the elderly in facing different physical conditions, economic conditions, and life situations [34].”

  1. In the revised version of the manuscript you explain that the total score of economic accessibility is the sum of the scores of the four separate questions. I understand the reasoning for calculating a sum score, but I do not understand the reasoning for recoding the sum scores ranging from 4-20 afterwards into values ranging from 1-5. Please justify this recoding procedure. Additionally, please correct the verb “recorded” and replace it by “recoded” as this might be what you have done, I guess.

Answer: According to the average segmentation method, According to the average segmentation method, the total score is the sum of the scores of the four questions, and the total score of 4-6 is recorded as 1, 7-10 is recorded as 2, 11-14 is recorded as 3, and 15-17 is recorded as 4,18 -20 is recoded as 5.

And correct the verb “recorded” and replace it by “recoded”.

  1. I see a contradiction with regard to the coding of residence and the interpretation of the results afterwards. In Table 1 it is stated that urban=1 and rural=2. In line 237 on p.6 you state that urban=1 and rural=0. This is contradictory and misleading. The coding of residence is important, however to properly be able to interpret the sign of the negative regression coefficient in Table 2. You write the following in the revised version of the manuscript: “The impact of residence on realized utilization is -0.2537 and it is significant, indicating that urban elderly people are not dependent on healthcare services and utilization frequency is lower.” (see line 292-294 on p.8). You should check the interpretation of the negative regression coefficient with regard to the proper coding of residence. Please elaborate on this.

Answer: In Table 1, urban=1 and rural=2 only represents two options, not real data. Urban=1 and rural=0 are dummy variables as empirical data. In table 1, increase the data in options column.

The impact of residence on realized utilization is -0.2537 and it is significant, indicating that urban elderly people are not dependent on community healthcare services and utilization frequency is lower. The possible reason is that the urban older adults have more opportunities to choose different types of healthcare services.

  1. The data collection is the result of a commendable effort, but besides a thorough description of the sampling procedure in the revised version of the manuscript, it is not well explained how the data collection took place in detail. Was it in form of personal interviews or in form of self-administered questionnaires? Did you deliver any incentives? How many respondents were excluded from analysis due to which reasons?

Answer: In 3.1, increase “Since the remuneration of a valid questionnaire is RMB 30, under the premise of unified training, every members carefully fills out the questionnaire in form of self-administered questionnaires, which ensuring the quality of the questionnaire to a certain extent”.

Finally, the team obtained a total of 948 valid questionnaires, and we adopt 832 valid questionnaires related to the theme, by discarding missing data and invalid questionnaires

  1. Additionally, as I have already mentioned, a native speaker should once more read through the whole manuscript to smooth out some remaining mistakes in grammar (e.g. lines 119-121 on p.3) and writing style.

Answer:  change the wrong sentence to “in China, the supply of healthcare services is at an early stage. The contents of healthcare services mostly focus on daily care, which are single and the quality is low.”